# Posttraumatic Osteomyelitis Risks Associated with *NLRP3* Gene Polymorphisms in the Chinese Population

**DOI:** 10.3390/jpm13020253

**Published:** 2023-01-30

**Authors:** Yudun Qu, Jiaxuan Li, Wei Zhang, Changliang Xia, Shuanji Ou, Yang Yang, Nan Jiang, Yunfei Ma, Yong Qi, Changpeng Xu

**Affiliations:** 1Department of Orthopaedics, Guangdong Second Provincial General Hospital, The Second School of Clinical Medicine, Southern Medical University, Guangzhou 510317, China; 2Department of Orthopaedics, Guangdong Second Provincial General Hospital, NO.466 Xingang Road, Haizhu District, Guangzhou 510317, China; 3Department of Orthopaedics, Nanfang Hospital, Southern Medical University, Guangzhou 510445, China

**Keywords:** single-nucleotide polymorphisms, posttraumatic osteomyelitis, rs10754558, rs7525979, case–control study

## Abstract

The purpose of this case–control study was to examine possible links between *NLRP3* gene polymorphisms and the risk of developing posttraumatic osteomyelitis (PTOM) in the Chinese population. A total of 306 patients with PTOM and 368 normal controls were genotyped for *NLRP3* (rs35829419, rs10754558, rs7525979, rs4612666), *ELP2* (rs1785929, rs1789547, rs1785928, rs12185396, rs681757, rs8299, rs2032206, rs559289), *STAT3* (rs4796793, rs744166, rs1026916, rs2293152, rs1053004), *CASP1* (rs501192, rs580253, rs556205, rs530537), NFKBIA (rs696), *NFKB1* (rs4648068), *CARD8* (rs204321), and *CD14* (rs2569190) using the genotyping technique SNaPshot. The genotype distributions of *NLRP3* gene rs10754558 (*p* = 0.047) and rs7525979 (*p* = 0.048) significantly differed between the patients and the healthy controls. Additionally, heterozygous models indicated a significant association between *NLRP3* rs10754558 and the likelihood of developing PTOM (OR = 1.600, *p* = 0.039), as did recessive and homozygous models of *NLRP3* rs7525979 (OR = 0.248, *p* = 0.019 and 0.239, *p* = 0.016, respectively). Collectively, our findings suggest that, in the Chinese population, the risk of developing PTOM was increased by the association between *NLRP3* rs10754558 and rs7525979. Therefore, our findings may provide novel insights and guidance in the prevention and development of PTOM.

## 1. Introduction

The likelihood of developing post-traumatic osteomyelitis (PTOM), a bone marrow infection with or without soft tissue infection, after trauma or orthopedic surgery, ranges from 0.4–16.1% [1], with incidences of 1–2% for closed fractures and approximately 30% for open fractures [2], with the highest incidence being 55% [3]. The diagnosis and treatment of PTOM are challenging owing to its “wide range” or “highly diverse” characteristics. This implies that the clinical presentation of each patient is unique. Furthermore, the clinical effect and prognosis are influenced by several factors, including the location of the bone, the extent and duration of the infection, the type and virulence of the pathogen, and the treatment strategy. Patients experience substantial physical, psychological [4], and socioeconomic stress [5] owing to prolonged illness and an increased risk of recurrent infections [6]. Therefore, reducing morbidity is as crucial as increasing healing rates. Therefore, a comprehensive understanding of PTOM pathogenesis, which is related to environmental and host factors, is essential.

Genetic predisposition plays a significant role in the development of PTOM. According to Wang et al., the GG genotype of the cyclooxygenase-2 (*COX-2*) gene enhances vulnerability to PTOM in the Chinese population [7]. Alves De Souza et al. reported that the Brazilian population might be more likely to acquire PTOM owing to interleukin (IL) gene polymorphisms rs16944 and rs2234663 [8]. These findings indicate that single-nucleotide polymorphisms (SNPs) contribute to PTOM onset.

The NOD-like receptor thermal protein domain-associated protein 3, also called *NLRP3*, or cryopyrin, is present on chromosome 1q44 and contains nine exons [9]. Granulocytes, monocytes, dendritic cells, T cells, B cells, epithelial cells, and osteoblasts are critical cell types that express cytoplasmic proteins [10]. Upon detecting tissue damage, *NLRP3* inflammasomes result in interleukin 1b (IL-1β), caspase 1 (Casp1)-mediated proteolysis, and interleukin-18 (IL-18) processing and in secretion, activation, and caspase 1 (Casp1)-mediated proteolysis [11]. In addition to neurological illnesses such as Parkinson’s disease and Alzheimer’s disease, NLRP3 signaling may play a role in various chronic and metabolic diseases, including type 2 diabetes, gout, inflammatory bowel disease, obesity, atherosclerosis, and periodic fever [12]. This is believed to be the root cause of degenerative disorders. However, studies on the relationship between PTOM risk and *NLRP3* SNPs, particularly in the Chinese population, are limited. Therefore, the objective of the current study was to evaluate any possible associations between PTOM susceptibility in the Chinese population and 25 SNPs associated with inflammation. Our findings may provide novel insights and guidance in the prevention and development of PTOM.

## 2. Materials and Methods

### 2.1. Patients and Controls

A total of 627 patients with chronic OM, 306 of whom had PTOM—which is defined as orthopedic surgery or trauma-related bone infection, accompanied by or without soft tissue infection—were enrolled between August 2016 to October 2019 [13] (Figure 1). One of the following confirmation criteria was used to determine the diagnosis of PTOM [14]: (1) a histopathological examination showing infection, (2) a clean sinus or fistula that connects to the implant or bone immediately, and (3) two or more cultures displaying the same pathogen. The patient group contained 251 men and 55 women with a median age was 44 years with an interquartile range (IQR) between 31.00–54.00. A total of 368 healthy controls, including 268 males and 100 females with a median age of 46 years and IQR between 37.00–52.00, were enrolled. The patient and control groups were matched in terms of age and sex after the adjustment (*p* > 0.05). The research methodology was approved by the hospital’s medical ethics committee, and each participant or their legal guardian completed an informed consent form on their behalf (approval letter No. NFEC-2019-087).

### 2.2. Determination of NLRP3 Polymorphisms 

Approximately 2 mL of peripheral blood from each patient was collected into ethylenediaminetetraacetic acid (EDTA) tubes. Genomic DNA was extracted from leukocytes using the salting-out method [15] and stored at −80 °C until further use. The Multiplex SNaPshot method [16] is a primer extension-based method for genotyping known SNP positions through the manuscript-automated DNA analyzer invented by Applied Biosystems, which was used for genotyping 25 tag SNPs in *NLRP3*, *ELP2*, *STAT3*, *CASP1*, *NFKBIA*, *NFKB1*, *CARD8*, and *CD14* (Applied Biosystems, Foster City, CA, USA). Appendix A lists the forward (F), reverse (R), and extension primers used in the PCR and extension procedures [17]. The detailed protocols have been previously described [14].

### 2.3. Outcome Parameters

The outcome criteria included the genotype distribution of 25 SNPs in PTOM patients and normal controls, mutation allele frequencies, and four genetic models (homozygous, heterozygous, dominant, and recessive). Differences were also observed in the levels of tumor necrosis factor-α (TNF-α) and interleukin-6 (IL-6) in the blood of individuals with abnormal genotypes.

### 2.4. Statistical Analysis

Statistical Product and Service Solutions (SPSS) program Version 13.0 was used for statistical analysis (SPSS Inc., Chicago, IL, USA). The Kolmogorov–Smirnov test was used to determine the normality of the data distribution. Continuous variables were reported as mean, standard deviation (SD), or median with IQR based on the data distribution. For comparisons between two distinct groups or between two or more groups, normally distributed data were compared using Student’s *t*-test or one-way analysis of variance (ANOVA). Otherwise, the Kruskal–Wallis H test or Mann–Whitney U test was used.

The chi-square test was used to determine whether the genotype distribution of healthy controls supported the Hardy–Weinberg equilibrium (HWE). The genotype distribution and frequency of mutant alleles in patients and healthy controls were compared using chi-square or Fisher’s exact test. To examine the potential relationship between genetic polymorphisms and the risk of developing PTOM according to four genetic models (dominant, recessive, homozygous, and heterozygous models), binomial logistic regression analysis with sex, age, and genotype distribution as covariates was used (CI). All reported *p*-values were two-sided, and statistical significance was set at *p* < 0.05.

## 3. Results

### 3.1. Clinical Characteristics of Patients with PTOM

Road traffic accidents were the most frequent cause of injury (46%), followed by blunt injuries (31%) and injuries from falls (12%). The tibia (51%), femur (30%), and calcaneus (7%) comprised the top three bones with open fractures, constituting two-thirds of all fractures. *Staphylococcus aureus* accounted for 32% of all pathogen detections, whereas the total proportion of positive intraoperative specimen cultures was 62%. The distribution of infection sites in the 306 patients with traumatic osteomyelitis is shown in Figure 1.

### 3.2. Results of HWE Tests 

In healthy controls, all 25 genotyped SNPs exhibited HWE balance. *p* > 0.05 indicates the genetic balance of the population; otherwise, it will increase the possibility of sampling bias (Appendix A).

### 3.3. Relationship between 25 SNPs and PTOM Susceptibility

#### 3.3.1. *NLRP3* rs10754558 and rs7525979 

The genotype distribution of *NLRP3* rs10754558 significantly differed between patients and healthy controls (*p* = 0.047). A further comparison revealed a significant association between rs10754558 and the risk of developing PTOM in the heterozygous (OR = 1.600, 95% CI: 1.021–2.507, *p* = 0.039) model, implying that the population with the CG genotype has a higher risk of developing PTOM. A positive association between *NLRP3* rs7525979 and PTOM susceptibility was identified in the recessive (OR = 0.248, 95% CI: 0.071–0.872, *p* = 0.019) and homozygous (OR = 0.239, 95% CI: 0.068–0.844, *p* = 0.016) models, indicating that the TT gene type appears to place the population at a higher risk of developing PTOM (Table 1).

#### 3.3.2. Serological IL-6 and TNF-α Levels in Various *NLRP3* rs10754558 and rs7525979 Patient Genotypes

The various genotypes of *NLRP3* rs10754558 and rs7525979 did not significantly vary in median serological IL-6 levels (*p* = 0.217 and *p* = 0.879, respectively; Table 2). Notably, PTOM patients with the rs10754558 CG genotype had median IL-6 levels higher than those with CC (*p* = 0.445), GG (*p* = 0.146), and GG + CC (*p* = 0.213; Figure 2) genotypes. Patients with genotypes CC (*p* = 0.053), GG (*p* = 0.065), and GG + CC (*p* = 0.014) had higher median TNF-α levels (Figure 2) compared with those of the CG genotype. Patients with the TT genotype of rs7525979 had higher IL-6 levels than those with the CT, CC, or CT + CC genotypes (*p* = 0.906, *p* = 0.953, and *p* = 0.996, respectively); however, the difference was not statistically significant (Figure 3). The median TNF-α levels of PTOM patients with the TT genotype of rs7525979 were higher than those of patients with the CT (*p* = 0.663), CC (*p* = 0.461), and CT + CC (*p* = 0.507) genotypes (Figure 3). Summarizing the research on the susceptibility of *NLRP3* highlighted its crucial role in inflammatory diseases (Table 3).

#### 3.3.3. Lack of Association between the Other 21 SNPs with PTOM Susceptibility 

Using this case–control study as evidence, we could not find any association between the other 21 SNPs and PTOM susceptibility in the Chinese population (*p* > 0.05; Appendix A). 

## 4. Discussion 

In recent years, it has become more evident that genetic predisposition is also implicated in the etiology of PTOM, with SNPs playing a significant role. We hypothesized that there might be a connection between SNPs in *NLRP3* and PTOM development in light of the significance of *NLRP3* in the inflammatory response and the well-established evidence that these SNPs are linked to the emergence of several other inflammatory disorders. Concordantly, the results of this study suggest that *NLRP3* rs10754558 and rs7525979 polymorphisms may be associated with increased susceptibility to PTOM in the Chinese population. Individuals with the CG genotype of rs10754558 and the TT genotype of rs7525979 may be at a high risk of PTOM. Furthermore, patients with the rs10754558 and rs7525979 genotypes at a higher risk of developing PTOM also had higher serum levels of IL-6 or TNF-α. The other 21 SNPs were not associated with susceptibility to PTOM in the Chinese population.

To reveal the relationship between *NLRP3* and inflammatory diseases, we summarized the research on the correlation between *NLRP3* and inflammatory diseases in the past decade (Table 3). PTOM development is influenced by external and host variables, as has already been established. Extrinsic variables, including the type and severity of the trauma, the position of the bone, the state of the soft tissues, the type and pathogenicity of the pathogen, and an early treatment plan, directly impact PTOM development. Intrinsic as well as environmental variables are crucial for PTOM development. Most earlier investigations examined the possible impact of host characteristics on lifestyle (such as obesity, smoking, and drinking), immunological state (such as immunocompromised status or immunodeficiency), and systemic and partial consequences (such as diabetes, anemia, cancer, venous stasis, and persistent lymphedema) [3,13,18,19,20]. 

*NLRP3* rs10754558 was correlated with an increased risk of developing PTOM, and the CG genotype was identified as a risk factor for PTOM for the first time. In a case–control study, Zhou et al. reported that the G allele of the *NLRP3* rs10754558 polymorphism was related to the onset of coronary artery disease in the Chinese population [21]. The GG and CG genotypes of *NLRP3* rs10754558 were strongly related to ulcerative colitis [22]. Moreover, CC genotype holders at the *NLRP3* rs10754558 locus may have a more severe systemic inflammatory response [23]. Addobbati et al. found that NLRP3 gene rs10754558 is vital in the development and onset of rheumatoid arthritis [24]. This is the first study to report that it plays a role in the development of rheumatoid arthritis. Although the genotype distribution differs across studies, it is likely affected by sample size, ethnicity, and the cause of the infection. In the current study, the median blood IL-6 levels of PTOM patients with the CG genotype were higher than those with other genotypes. This suggests that rs10754558 is implicated in PTOM via the control of IL-6 levels.

The TT genotype might be a risk factor for the association of the *NLRP3* variant rs7525979 with PTOM susceptibility. Based on the experimental findings of this study, there is a strong link between them, even if prior studies have not offered pertinent data. The genotype distribution, mutant allele frequency, and heterozygous model *p*-values between the patient and normal groups were all <0.05, suggesting that the variant allele T may be a risk factor for PTOM, and the TT genotype may be a high-risk population for PTOM despite the lack of significantly different IL-6 levels. This may be attributed to the different periods in which serological indicators were measured in the selected patients. To obtain more comprehensive findings, investigations with larger sample sizes are required. The median IL-6 levels of patients with rs7525979 genotype TT were higher than those of patients with other genotypes, suggesting a similar mechanism to that of rs10754558.

The two SNPs (*NLRP3* rs10754558 and rs7525979) substantially linked to PTOM development in this Chinese population were all identified in promoter regions, resulting in altered expression levels of their respective proteins. This may play a role in the genesis of PTOM. These two SNPs may affect the effectiveness of PTOM therapy, in addition to being implicated in PTOM pathophysiology. Future research is required to ascertain whether SNPs also affect the risk of infection recurrence given that its recurrence rate after bone infection is reportedly higher than 20–30% [25,26]. One approach is to utilize these SNPs to predict the likelihood of PTOM in certain individuals, as the current investigation found that SNPs in two *NLRP3* genes may be implicated in PTOM development. However, before this application can be achieved, at least two necessary steps must be undertaken. First, the sample size must be increased to obtain conclusion information regarding the association between such SNPs and the likelihood of PTOM development. Second, cohort studies must be conducted to comprehensively evaluate the predictive power of such SNPs for PTOM.

*NLRP3* rs35829419 and rs4612666 were not statistically different from PTOM in genotype distribution, mutant allele frequency, the homozygous/heterozygous model, or IL-6 and TNF-α levels, indicating that rs35829419 and rs4612666 may not be associated with the occurrence and development of PTOM. In the remaining seven genes, some studies have found that *STAT3* (rs4796793, rs744166, rs1026916, rs2293152, rs1053004) is involved in the regulation of inflammatory osteolysis in bone infection [27], and the level of *CASP-1* (rs501192, rs580253, rs556205, rs530537) in peripheral blood mononuclear cells of freshly isolated multifocal osteomyelitis patients in active and remission stages is significantly higher than that of healthy controls [28], suggesting that *STAT3* and *CASP-1* may be the risk genes for PTOM. In our research, we did not find that *ELP2* (rs1785929, rs178989547, rs1785928, rs12185396, rs681757, rs8299, rs2032206, rs559289), *NFKBIA* (rs696), *NFKB1* (rs4648068), *CARD8* (rs204321), or *CD14* (rs2569190) were associated with the risk of PTOM in Chinese populations. We also did not find evidence that *ELP2, NFKBIA, NFKB1, CARD8,* or *CD14* were associated with PTOM in previous studies.

This research also had several limitations. First, the sample size was small because this was a genome-wide association study (GWAS). It is challenging to include suitable study participants in a limited time, and future studies will extend to a wider population with a control group of other ethnic origins, such as whites, blacks, and other minorities. Second, the relationship between the different *NLRP3* rs10754558 and rs7525979 genotypes remains unclear because we could not discover any serological proof of IL-1β and IL-18. Third, the exact processes of *NLRP3* rs35829419 and rs4612666 in the pathogenesis of PTOM must be investigated further, as this study is only a preliminary report. Finally, the detailed mechanisms of the two SNPs in PTOM pathogenesis should be further investigated. Currently, the incidence of chronic osteomyelitis, especially traumatic osteomyelitis, is gradually increasing, and the efficacy of current treatment methods remains unclear. Therefore, the risk of postoperative infection remains high. Therefore, our findings may provide novel insights and guidance in the prevention and development of PTOM. 

In conclusion, the GG genotype for rs10754558 and the TT genotype for rs7525979 of the *NLRP3* were associated with elevated odds of developing PTOM in this Chinese cohort. 

## Figures and Tables

**Figure 1 jpm-13-00253-f001:**
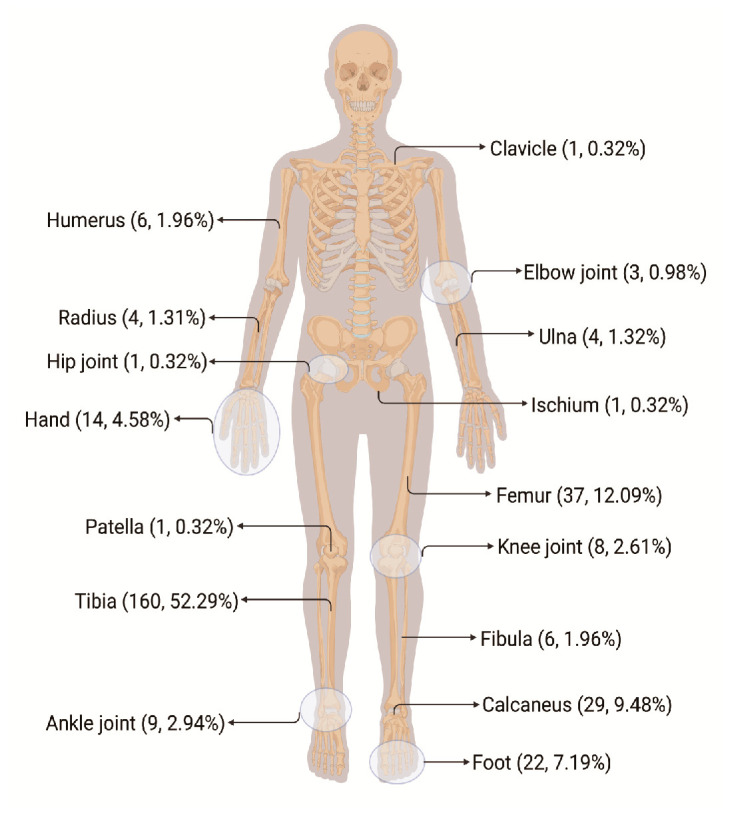
Distribution map of single infection sites in 306 patients with traumatic osteomyelitis.

**Figure 2 jpm-13-00253-f002:**
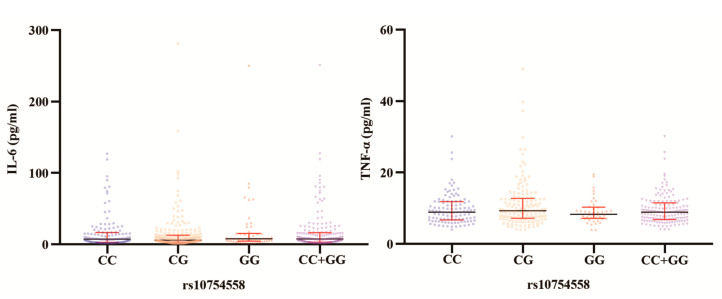
Serological IL-6 and TNF-α levels among different rs10754558 genotypes in PTOM patients.

**Figure 3 jpm-13-00253-f003:**
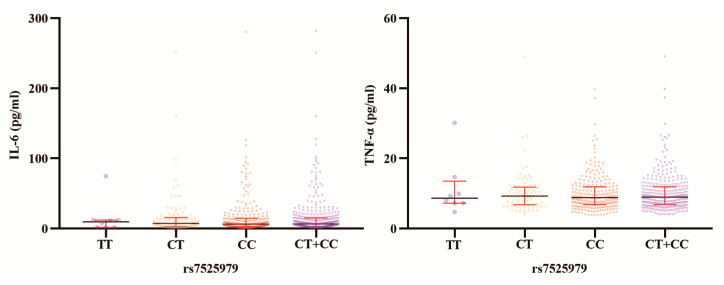
Serological IL-6 and TNF-α levels among different rs7525979 genotypes in PTOM patients.

**Table 1 jpm-13-00253-t001:** Comparisons of genotype distribution, allele frequency, and genetic models of rs35829419, rs10754558, rs7525979, and rs4612666 between PTOM patients and healthy controls.

SNP	Item	Allele or Genotype	Patients	Controls	*p*-Values	OR (95% CI)
rs35829419	Genotype (n, %)	AA	0 (0.0)	0 (0.0)	NA	NA
		AC	2 (0.65)	1 (0.27)		
		CC	304 (99.35)	367 (99.73)		
	Allele frequency	A vs. C	2/610	1/735	0.873	2.410 (0.218, 26.640)
	Dominant model	AA + AC vs. CC	2/304	1/367	0.873	2.414 (0.218, 26.756)
	Recessive model	AA vs. AC + CC	0/306	0/368	NA	NA
	Homozygous model	AA vs. CC	0/304	0/367	NA	NA
	Heterozygous model	AC vs. CC	2/304	1/367	0.873	2.414 (0.218, 26.756)
rs10754558	Genotype (n, %)	CC	97 (31.70)	135 (36.68)	0.047	NA
		CG	169 (55.23)	169 (45.92)		
		GG	40 (13.07)	64 (17.39)		
	Allele frequency	C vs. G	363/249	439/297	0.901	0.986 (0.793, 1.227)
	Dominant model	CC + CG vs. GG	266/40	304/64	0.122	1.400 (0.913, 2.148)
	Recessive model	CC vs. CG + GG	97/209	135/233	0.175	0.801 (0.581, 1.104)
	Homozygous model	CC vs.GG	97/40	135/64	0.564	1.150 (0.716, 1.846)
	Heterozygous model	CG vs.GG	169/40	169/64	0.039	1.600 (1.021, 2.507)
rs7525979	Genotype (n, %)	TT	3 (0.98)	14 (3.84)	0.048	NA
		CT	80 (26.14)	102 (27.95)		
		CC	223 (72.88)	249 (68.22)		
	Allele frequency	T vs. C	86/526	130/600	0.062	0.755 (0.561, 1.015)
	Dominant model	TT + CT vs. CC	83/223	116/249	0.188	0.799 (0.572, 1.117)
	Recessive model	TT vs. CT + CC	3/303	14/351	0.019	0.248 (0.071, 0.872)
	Homozygous model	TT vs. CC	3/223	14/249	0.016	0.239 (0.068, 0.844)
	Heterozygous model	CT vs. CC	80/223	102/249	0.450	0.876 (0.621, 1.235)
rs4612666	Genotype (n, %)	CC	73 (23.86)	98 (26.63)	0.265	NA
		CT	168 (54.90)	179 (48.64)		
		TT	65 (21.24)	91 (24.73)		
	Allele frequency	C vs. T	314/298	375/361	0.896	1.014 (0.819, 1.257)
	Dominant model	CC + CT vs. TT	241/65	277/91	0.285	1.218 (0.848, 1.750)
	Recessive model	CC vs. CT + TT	73/233	98/270	0.401	0.863 (0.608, 1.225)
	Homozygous model	CC vs. TT	73/65	98/91	0.852	1.043 (0.672, 1.618)
	Heterozygous model	CT vs. TT	168/65	179/91	0.160	1.314 (0.897, 1.925)

PTOM: posttraumatic osteomyelitis; OR: odds ratio; CI: confidence interval.

**Table 2 jpm-13-00253-t002:** Serological levels of WBC, CRP, PCT IL-6, TNF-α, and SAA among different genotypes of rs35829419, rs10754558, rs7525979, and rs4612666 in the patient group.

Items	rs35829419				rs10754558				rs7525979				rs4612666			
	AA	AC	CC	*p* Value	CC	CG	GG	*p* Value	TT	CT	CC	*p* Value	CC	CT	TT	*p* Value
WBC (×10/L)^9^ Median (IQR)	NA	6.065 (5.460, 6.065)	6.980 (5.720, 8.320)	0.365	7.03 (5.72, 8.61)	6.96 (5.63, 8.25)	6.82 (5.62, 8.30)	0.917	6.35 (5.88, 6.35)	7.44 (5.75, 8.23)	6.81 (5.66, 8.37)	0.711	6.79 (5.52, 8.82)	6.99 (5.86, 8.34)	6.86 (5.68, 8.00)	0.624
CRP (mg/L) Median (IQR)	NA	5.840 (2.100, 5.840)	4.345 (1.485, 11.920)	0.870	5.61 (1.73, 15.24)	4.09 (1.40, 9.83)	3.25 (1.42, 8.10)	0.282	8.70 (7.10, 8.70)	5.30 (1.67, 15.24)	4.05 (1.42, 9.98)	0.160	3.00 (0.98, 8.61)	4.58 (1.99, 12.47)	4.03 (1.55, 16.42)	0.129
ESR (mm/h) Median (IQR)	NA	7.000 (1.000, 7.000)	15.000 (7.000, 36.000)	0.184	20.00 (7.75, 42.50)	14.00 (7.00, 32.00)	13.00 (4.25, 28.25)	0.399	36.00 (5.00, 36.00)	14.00 (6.00, 43.00)	16.00 (7.00, 32.00)	0.698	13.00 (5.00, 22.75)	16.00 (7.00, 39.50)	15.00 (7.75, 44.00)	0.257
PCT (ng/mL) Median (IQR)	NA	NA	0.037 (0.021, 0.060)	NA	0.03 (0.02, 0.05)	0.04 (0.03, 0.07)	0.03 (0.02, 0.04)	0.258	0.03 (0.03, 0.03)	0.04 (0.03, 0.05)	0.03 (0.02, 0.06)	0.488	0.03 (0.02, 0.06)	0.04 (0.02, 0.07)	0.03 (0.02, 0.05)	0.585
IL-6 (pg/mL) Median (IQR)	NA	7.660 (6.010, 7.660)	5.270 (2.895, 10.930))	0.518	6.83 (2.81, 14.49)	4.67 (2.75, 9.65)	5.86 (3.18, 8.51)	0.217	12.16 (1.54, 12.16)	5.71 (2.50, 11.76)	5.05 (3.00, 9.89)	0.879	4.72 (2.98, 8.32)	5.61 (2.94, 11.19)	5.60 (2.39, 14.41)	0.455
TNF-α (pg/mL) Median (IQR)	NA	7.015 (6.150, 7.015)	8.320 (6.570, 10.800)	0.394	8.55 (6.43, 10.80)	8.31 (6.50, 11.05)	8.06 (6.93, 9.77)	0.855	8.02 (7.23, 8.02)	9.08 (6.78, 11.33)	8.08 (6.49, 10.73)	0.679	7.71 (6.71, 10.10)	8.88 (6.74, 11.50)	7.38 (6.29, 10.70)	0.526
SAA (mg/L) Median (IQR)	NA	12.750 (6.800, 12.750)	11.400 (6.400, 35.900)	0.880	12.70 (6.70, 51.55)	10.90 (6.20, 33.18)	10.90 (6.85, 24.30)	0.818	35.90 (12.70, 35.90)	12.65 (7.38, 56.70)	10.95 (6.20, 32.70)	0.296	8.85 (5.88, 23.75)	12.10 (6.70, 36.10)	13.05 (6.95, 67.28)	0.174

**Table 3 jpm-13-00253-t003:** Diseases associated with NLRP3 gene susceptibility.

Number	PubMed ID	SNP	Related Diseases
1	33848452 30658261	Q705K	Cystic fibrosis Multiple sclerosis
2	31978095 28116820	rs4612666	Periodontitis Ankylosing spondylitis
3	33533549	rs3806265	Myasthenia gravis
4	29888470 26689701	rs10754558	Acne vulgaris Ischemic stroke
5	30131971	rs7525979	Parkinson’s disease
6	23588528	rs7512998	Primary gouty arthritis
7	29850521	rs3806265, rs10754557	Psoriasis vulgaris
8	21621776	rs35829419	Abdominal aortic aneurysms
9	22112657	rs2027432, rs12048215	MOD syndrome
The present study	NA	Rs10754558, rs7525979	PTOM

SNP: single-nucleotide polymorphisms.

## Data Availability

The datasets generated and/or analyzed during the current study are not publicly available because of the respect and protection of the privacy of the patients but are available from the corresponding author upon reasonable request.

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
