# Peer review of "Posttraumatic Osteomyelitis Risks Associated with NLRP3 Gene Polymorphisms in the Chinese Population"

_jpm, 2023, doi:10.3390/jpm13020253_

Round 1

Reviewer 1 Report

A thorough revision of the design and the writing is required, with the addition of a geneticist who is experienced in associations studies. There are major issues with the design of the study; 1)There is no description of the characteristics and matching of controls, 2)The examined SNPs are distributed over 8 genes, yet everything discusses NLRP3 gene polymorphisms, 3) The number of cases and controls is small. There are writing issues, such as; There are sentences that contained confusing language such as "salting the leukemic cells," "four genetic models" but list only 3, and "Multiplex SNaPshot method." 

Author Response

Response to review #1:

Comments and Suggestions for Authors:

A thorough revision of the design and the writing is required, with the addition of a geneticist who is experienced in associations studies. There are major issues with the design of the study; 1) There is no description of the characteristics and matching of controls, 2) The examined SNPs are distributed over 8 genes, yet everything discusses NLRP3 gene polymorphisms, 3) The number of cases and controls is small. There are writing issues, such as; There are sentences that contained confusing language such as "salting the leukemic cells," "four genetic models" but list only 3, and "Multiplex SNaPshot method." 

Comment 1:A thorough revision of the design and the writing is required, with the addition of a geneticist who is experienced in associations studies.

Response 1.1: Thank you for your kind comments.

Based on your suggestions, I have thorough revised the design and writing of this study. We adjusted the narrative and logic of the article, polished the writing, and added a description of experimental details and relevant citations in the methods section (as following). The modified sections are highlighted in red in the article.

  1. 2 Determination of NLRP3 polymorphisms

Approximately 2 mL of peripheral blood from each patient was collected into ethylenediaminetetraacetic acid (EDTA) tubes. Genomic DNA was extracted from leukocytes using the salting-out method [15] and stored at −80°C until further use. The Multiplex SNaPshot method [16] is a primer extension-based method for genotyping known SNP positions through the manuscript automated DNA analyzer invented by Applied Biosystems, which was used for genotyping 25 tag SNPs in NLRP3, ELP2, STAT3, CASP1, NFKBIA, NFKB1, CARD8, and CD14 (Applied Biosystems, Foster City, USA).Table S1 lists the forward (F), reverse (R), and extension primers used in the PCR and extension procedures [17]. The detailed protocols have been previously described [14].

Response 1.2:

Dr. Nan Jiang from Nanfang Hospital, Southern Medical University, who had published several genetic studies (as following), promoted the progress of this study and guided the experimental operation. According to your comments, Dr. Nan Jiang re-evaluated the study.

[1] Zhao XQ, Chen K, Wan HY, et al. Vitamin D Receptor Genetic Variations May Associate with the Risk of Developing Late Fracture-Related Infection in the Chinese Han Population. J Immunol Res. 2022;2022: 9025354. Published 2022 Feb 10. doi:10.1155/2022/9025354

[2] Zhao XQ, Wan HY, He SY, Qin HJ, Yu B, Jiang N. Vitamin D Receptor Genetic Polymorphisms Associate With a Decreased Susceptibility to Extremity Osteomyelitis Partly by Inhibiting Macrophage Apoptosis Through Inhibition of Excessive ROS Production via VDR-Bmi1 Signaling. Front Physiol. 2022; 13:808272. Published 2022 Jul 25. doi:10.3389/fphys.2022.808272

Comment 2:There is no description of the characteristics and matching of controls.

Response: Thank you for your precious comments.

Based on your comments, we described the characteristics of the control group in greater detail, adding medians and interquartile ranges for age, and comparing them with patient group (as following). The case and control groups were age matched, while the sex ratio was statistically significant, but the results were not different after adjusting for sex ratio.

2.1 Patients and controls

A total of 627 patients with chronic OM, 306 of whom had PTOM, which is defined as orthopedic surgery or trauma-related bone infection, accompanied by or without soft tissue infection, were enrolled between August 2016 to October 2019 [13] (Fig 1). One of the following confirmation criteria was used to determine the diagnosis of PTOM [14]: 1) a histopathological examination showing infection, 2) a clean sinus or fistula that connects to the implant or bone immediately, and 3) two or more cultures displaying the same pathogen. The patient group contained 251 men and 55 women with a median age was 44 years with an interquartile range (IQR) between 31.00–54.00. A total of 368 healthy controls, including 268 males and 100 females with a median age of 46 years and IQR between 37.00–52.00, were enrolled. The patient and control groups were matched in terms of age and sex after the adjustment (P > 0.05). The research methodology was approved by the hospital's medical ethics committee, and each participant completed an informed consent form on behalf of themselves or their legal guardians. Approval letter No. NFEC-2019-087.

Comment 3: The examined SNPs are distributed over 8 genes, yet everything discusses NLRP3 gene polymorphisms.

Response: Thank you for your precious comments.

Since the correlation analysis of ELP2, STAT3, CASP1, NFKBIA, NFKB1, CARD8, and CD14 was not statistically significant in this study, we did not discuss it too much. According to your comment, we have added a discussion of the above 7 genes in the discussion section. See paragraph 6 of the discussion or as following.

The NLRP3 rs35829419 and rs4612666 were not statistically different from PTOM in genotype distribution, mutant allele frequency, homozygous/heterozygous model, and IL-6 and TNF-α levels, indicating that rs35829419 and rs4612666 may not be associated with the occurrence and development of PTOM. In the remaining 7 genes, some studies have found that STAT3 (rs4796793, rs744166, rs1026916, rs2293152, rs1053004) is involved in the regulation of inflammatory osteolysis in bone infection [27], and the level of CASP-1 (rs501192, rs580253, rs556205, rs530537) in peripheral blood mononuclear cells of freshly isolated multifocal osteomyelitis patients in active and remission stages is significantly higher than that of healthy controls [28], suggesting that STAT3 and CASP-1 may be the risk genes for PTOM. In our research, we did not find that ELP2 (rs1785929, rs178989547, rs1785928, rs12185396, rs681757, rs8299, rs2032206, rs559289), NFKBIA (rs696), NFKB1 (rs4648068), CARD8 (rs204321), or CD14 (rs2569190) were associated with the risk of PTOM in Chinese populations. And we also did not find evidence that ELP2, NFKBIA, NFKB1, CARD8 and CD14 were associated with PTOM in previous studies.

Comment 4: The number of cases and controls is small. 

Response: Thank you for your precious comments.

Yes, you are right that the small sample size is indeed a limitation of our study. We will emphasize this point in the limitation section. Larger sample sizes should be used in future investigations to verify the reliability of our study. See paragraph 6 of the discussion or as following.

This research also had several limitations. First, the sample size was small because this was a genome-wide association study (GWAS). It is challenging to include suitable study participants in a limited time, and future study will extend to a wider population with also a control group of other ethnic origin, such as whites, blacks and other minorities. Second, the relationship between the different NLRP3 rs10754558 and rs7525979 genotypes remains unclear because we could not discover any serological proof of IL-1β and IL-18. Third, the exact processes of NLRP3 rs35829419 and rs4612666 in the pathogenesis of PTOM must be investigated further, as this study is only a preliminary report. Finally, the detailed mechanisms of the two SNPs in PTOM pathogenesis should be further investigated. Currently, the incidence of chronic osteomyelitis, especially traumatic osteomyelitis, is gradually increasing, and the efficacy of current treatment methods remains unclear. Therefore, the risk of postoperative infection remains high. Therefore, our findings may provide novel insights and guidance in the prevention and development of PTOM.

Finally, thanks a lot again for your review on our manuscript and your insightful suggestions. We are looking forward to your further advice on this and above points.

Reviewer 2 Report

l  Authors should check and revise their manuscripts in accordance with JPM's Author Guidelines.

l  Xie Xiaodong, a professor at the Institute of Life Sciences at Lanzhou University in Gansu Province, China, is known to have explained, "There are no 'Han's' of pure descent, and DNA surveys show that the characteristics of various ethnic groups are evenly combined."

The author conducted a study on the risk of post-traumatic osteomyelitis related to the NLRP3 gene polymorphism of the Han people, and I wonder why this study was conducted only in the "Han people."

l  The text in figure 1 are too small and the resolution of the figure is too low to visually check the figures.

l  There is no specific description of the experimental method in Materials and Methods.

l  The Conclusion part was poorly written. The message is not clear.

Author Response

Response to review #2:

Comments to the Author:

Authors should check and revise their manuscripts in accordance with JPM's Author Guidelines.

Xie Xiaodong, a professor at the Institute of Life Sciences at Lanzhou University in Gansu Province, China, is known to have explained, "There are no 'Han's' of pure descent, and DNA surveys show that the characteristics of various ethnic groups are evenly combined." The author conducted a study on the risk of post-traumatic osteomyelitis related to the NLRP3 gene polymorphism of the Han people, and I wonder why this study was conducted only in the "Han people."

The text in figure 1 are too small and the resolution of the figure is too low to visually check the figures.

There is no specific description of the experimental method in Materials and Methods.

The conclusion part was poorly written. The message is not clear.

Comment 1:Authors should check and revise their manuscripts in accordance with JPM's Author Guidelines.

Response: Thanks for your suggestion.

We checked and revised our manuscripts in accordance with JPM’s Author Guidelines. We have added supplementary materials, acknowledgments Institutional Review Board Statement and Informed Consent Statement. We also reformatted the article and adjusted the citation format (see as following).

Supplementary Materials: Table S1: PCR primers and extension primers of the interleukin gene polymorphisms. Table S2: HWE test outcomes of the 25 gene polymorphisms for healthy controls. Table S3: Comparisons of genotype distribution, allele frequency, and genetic models of the remaining 21 SNPs between PTOM patients and healthy controls.

Acknowledgments: The authors thank all participants and staff who made this study possible.

Institutional Review Board Statement: Approval letter No. NFEC-2019-087.

Informed Consent Statement: Informed consent was obtained from all subjects involved in the study.

Comment 2: The author conducted a study on the risk of post-traumatic osteomyelitis related to the NLRP3 gene polymorphism of the Han people, and I wonder why this study was conducted only in the "Han people."

Response: Thanks for your precious suggestion.

According to your opinion, we have consulted relevant materials, and as Professor Xiaodong Xie said, there is no pure Han people. Therefore, we have revised the relevant narrative and changed the Chinese Han population to the Chinese population. Because we did not carry out further data mining, we did not have enough understanding of the gene polymorphism of the Chinese population, resulting in such an error. Thank you very much for your comments, and we will pay attention to this aspect in the future research.

Comment 3: The text in figure 1 are too small and the resolution of the figure is too low to visually check the figures.

Response: Thanks for your precious suggestion.

According to your opinion, we modified the text and image in Figure 1 to achieve higher definition.

Comment 4: There is no specific description of the experimental method in Materials and Methods.

Response: Thanks for your precious suggestion.

Based on your comments, we have added the description and citations of the experimental methods in the Methods section to increase the readability of the article. See paragraph 2 of Materials and Methods for details or as following.

  1. 2 Determination of NLRP3 polymorphisms

Approximately 2 mL of peripheral blood from each patient was collected into ethylenediaminetetraacetic acid (EDTA) tubes. Genomic DNA was extracted from leukocytes using the salting-out method [15] and stored at −80°C until further use. The Multiplex SNaPshot method [16] is a primer extension-based method for genotyping known SNP positions through the manuscript automated DNA analyzer invented by Applied Biosystems, which was used for genotyping 25 tag SNPs in NLRP3, ELP2, STAT3, CASP1, NFKBIA, NFKB1, CARD8, and CD14 (Applied Biosystems, Foster City, USA). Table S1 lists the forward (F), reverse (R), and extension primers used in the PCR and extension procedures [17]. The detailed protocols have been previously described [14].

Comment 5: The conclusion part was poorly written. The message is not clear.

Response: Thanks for your precious suggestion.

Your comments are very important. In view of your comments, we have re-adjusted the narrative and logical relationship, and polished the text to increase the accuracy and readability of the narrative.

Finally, we thank you again for your several suggestions for our study, which are very insightful. We are looking forward to your further advice on this and above points.

Reviewer 3 Report

The work is interesting, well conducted and with appropriate bibliographic citations. It would be interesting to extend the research to a wider population with also a control group of other ethnic origin

Author Response

Response to review #3:

Comments to the Author:

The work is interesting, well conducted and with appropriate bibliographic citations. It would be interesting to extend the research to a wider population with also a control group of other ethnic origin.

Comment 1:It would be interesting to extend the research to a wider population with also a control group of other ethnic origin.

Response: Thanks for your precious suggestion.

It is an honor to receive your recognition of our research. The small sample size of our study is indeed a limitation, which we highlight in our limitation. Future study will expand to a wider population, as well as controls of other ethnic ancestry, such as domestic minorities, and Caucasians and blacks in the world(see paragraph 7 of the discussion).

This research also had several limitations. First, the sample size was small because this was a genome-wide association study (GWAS). It is challenging to include suitable study participants in a limited time, and future study will extend to a wider population with also a control group of other ethnic origin, such as whites, blacks and other minorities. Second, the relationship between the different NLRP3 rs10754558 and rs7525979 genotypes remains unclear because we could not discover any serological proof of IL-1β and IL-18. Third, the exact processes of NLRP3 rs35829419 and rs4612666 in the pathogenesis of PTOM must be investigated further, as this study is only a preliminary report. Finally, the detailed mechanisms of the two SNPs in PTOM pathogenesis should be further investigated. Currently, the incidence of chronic osteomyelitis, especially traumatic osteomyelitis, is gradually increasing, and the efficacy of current treatment methods remains unclear. Therefore, the risk of postoperative infection remains high. Therefore, our findings may provide novel insights and guidance in the prevention and development of PTOM.

Finally, thank you again for your comments on the manuscript. We are looking forward to your further advice on this point.

Round 2

Reviewer 1 Report

Much improved manuscripts, minor edits in style and spelling are needed

Author Response

Response to review #1 (Round 2):

Comments and Suggestions for Authors:

Much improved manuscripts, and minor edits in style and spelling are needed.

Response: Thank you for your comments.

    Based on your suggestions, we made some changes to the article, including edits in style and spelling.

Finally, thanks a lot again for your review of our manuscript and your suggestions. We are looking forward to your further advice.

Reviewer 2 Report

The resolution in Figure 1 is still low. Please raise the resolution by increasing the font size.

The author's affiliation is not concise. Please check JBM's submission format. There is no need to mark MD or MM. Authors from the same affiliation should be grouped together.

Author Response

Response to review #1 (Round 2):

Comments and Suggestions for Authors:

  1. The resolution in Figure 1 is still low. Please raise the resolution by increasing the font size.
  2. The author's affiliation is not concise. Please check JBM's submission format. There is no need to mark MD or MM. Authors from the same affiliation should be grouped together.

Comment 1:The resolution in Figure 1 is still low. Please raise the resolution by increasing the font size.

Response 1: Thank you for your kind comments.

According to your suggestion, I have adjusted the clarity and increased the font size.

Comment 2: The author's affiliation is not concise. Please check JBM's submission format. There is no need to mark MD or MM. Authors from the same affiliation should be grouped together.

Response: Thank you for your precious comments.

Based on your comments, I modified the author's affiliation and removed unnecessary components according to JPM's submission format.

   Yu-Dun Qu 1, Jia-Xuan Li 1, Wei Zhang 3, Chang-Liang Xia 3, Shuan-Ji Ou 3, Yang Yang 3, Nan Jiang 2, Yun-Fei Ma 2, Yong Qi 3, Chang-Peng Xu 3

  1. Department of Orthopaedics, Guangdong Second Provincial General Hospital, The Second School of Clinical Medicine, Southern Medical University, Guangzhou, Guangdong, P.R. China; Email: [email protected] (Y-D.Q); [email protected] (J-X.L);
  2. Department of Orthopaedics, Nanfang Hospital, Southern Medical University, Guangzhou, Guangdong, P.R. China; Email: [email protected] (N.J); [email protected] (Y-F.M);
  3. Department of Orthopaedics, Guangdong Second Provincial General Hospital, Guangzhou, Guangdong, P.R. China; Email: [email protected] (W.Z); [email protected] (C-L.X); [email protected] (S-J.O); [email protected] (Y.Y); [email protected] (Y.Q); [email protected] (C-P.X).

Finally, thanks a lot again for your review of our manuscript and your insightful suggestions. We are looking forward to your further advice.
